# StealthDiffusion: Towards Evading Diffusion Forensic Detection through Diffusion Model

## ABSTRACT

The rapid progress in generative models has given rise to the critical task of AI-Generated Content Stealth (AIGC-S), which aims to create AI-generated images that can evade both forensic detectors and human inspection. This task is crucial for understanding the vulnerabilities of existing detection methods and developing more robust techniques. However, current adversarial attacks often introduce visible noise, have poor transferability, and fail to address spectral differences between AI-generated and genuine images. To address this, we propose StealthDiffusion, a framework based on stable diffusion that modifies AI-generated images into high-quality, imperceptible adversarial examples capable of evading state-of-the-art forensic detectors. StealthDiffusion comprises two main components: Latent Adversarial Optimization, which generates adversarial perturbations in the latent space of stable diffusion, and Control-VAE, a module that reduces spectral differences between the generated adversarial images and genuine images without affecting the original diffusion model's generation process. Extensive experiments demonstrate the effectiveness of StealthDiffusion in both white-box and black-box settings, transforming AI-generated images into higher-quality adversarial forgeries with frequency spectra resembling genuine images. These images are classified as genuine by state-of-the-art forensic classifiers and are difficult for humans to distinguish.

## CCS CONCEPTS

• **Security and privacy** → **Social aspects of security and privacy**; • **Computing methodologies** → **Computer vision**;

## KEYWORDS

Computer vision, AI-Generated image, Adversarial attacks

## 1 INTRODUCTION

In recent years, generative models, particularly diffusion-based image synthesis techniques [18], have made significant progress in deep learning and excelled at generating highly realistic images. As these generative technologies become increasingly widespread, it is crucial to develop techniques that can create AI-generated images indistinguishable from genuine ones by both human eyes and AI-based detectors. This will help identify the limitations and weaknesses of current detection methods and contribute to the

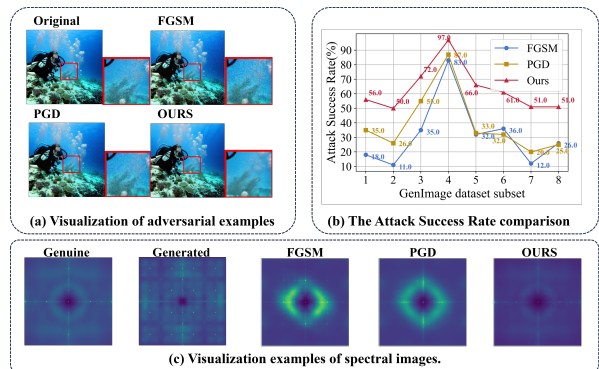

**Figure 1: Quantitative and qualitative comparison analysis:(a) Visual examples of spectral images comparing baseline methods and our method. The result of the baseline still contains visible artifacts, whereas the spectral images produced by our proposed method are most similar to the genuine images.(b) Visualization of adversarial examples generated by baseline methods and our method.Our method achieves higher image quality.(c) Quantitative performance comparison of baseline methods and our method on GenImage [49].**

development of more robust detection models. We refer to this as the *AI-Generated Content Stealth* (**AIGC-S**) task, which aims to generate AI-generated images that can evade detection by both human perception and AI-based algorithms. The goal of this task is to apply carefully crafted perturbations to existing AI-generated images, making them indistinguishable from genuine images while maintaining their visual quality. By achieving this, we can gain valuable insights into the vulnerabilities of current detection methods and develop more robust and reliable detection algorithms.

Recent approaches to improving image stealth against detection have primarily focused on adding adversarial noise directly at the image level. For example, the Fast Gradient Sign Method (FGSM) [15] creates noise by perturbing the image in the direction of the gradient of the loss with respect to the input image. Projected Gradient Descent (PGD) [30] iteratively applies small perturbations, making it a more powerful, though computationally expensive, approach. AutoAttack [7] is an ensemble of attacks that optimizes adversarial perturbations to test model robustness effectively. However, we argue that traditional attack methods have three main limitations when applied to the AIGC-S task: (1) These methods often introduce visible noise to diffusion-generated images, as shown in Fig. 1 (a), compromising the visual quality and failing to evade human perception. (2) Despite high success rates in white-box scenarios, their transferability to black-box settings is poor, with attack success rates of only 35.38% and 31.63% for PGD and FGSM, respectively, as illustrated in Fig. 1 (b). (3) These

methods only add noise in the spatial domain, ignoring the spectral differences between genuine and generated images. Previous studies [10, 13, 19–21, 23, 24, 26, 27, 29, 34, 43] have shown that spectral features are crucial for detection models to distinguish between genuine and generated images. Fig. 1 (c) demonstrates that the spectra of adversarial images generated by traditional FGSM and PGD methods differ significantly from those of genuine images, leading to suboptimal attack performance.

To address these limitations, we propose a novel approach called StealthDiffusion, which enhances the stealth of AI-generated images against detection by optimizing in the latent space and reducing spectral differences between generated and genuine images. Specifically, StealthDiffusion consists of two key components. The first component is Latent Adversarial Optimization (LAO), which harnesses the powerful generative and representational capabilities of Stable Diffusion to perform adversarial optimization in its latent space. By optimizing in this latent space, LAO enables more detailed and comprehensive image optimization, resulting in higher-quality stealth images. The second component is the Control-VAE module, which aims to minimize the spectral differences between generated and genuine images. It achieves this by reconstructing both genuine and generated images using a VAE model and then integrating this knowledge into the Stable Diffusion decoder through a control-net-like skip-connection method. This innovative approach effectively reduces spectral aliasing, making the generated images more indistinguishable from genuine ones in the spectral domain.

The effectiveness of StealthDiffusion is evident in Fig. 1, which showcases its advantages over traditional attack methods. From a visual perspective, Fig. 1 (a) demonstrates that StealthDiffusion generates higher-quality images without the perceptible noise artifacts that plague traditional methods. Moreover, Fig. 1 (b) highlights StealthDiffusion's superior transferability, as it outperforms traditional methods by 27.63% in challenging black-box transfer attacks. Lastly, Fig. 1 (c) reveals that the spectra of images processed by StealthDiffusion closely resemble those of genuine images, eliminating the telltale spectral forgery patterns. This is a testament to the Control-VAE module's effectiveness in bridging the spectral gap between generated and genuine images.

Our contributions can be summarized as follows:

- We are the first to focus on the detectability in diffusion-generated forged images, leading to a foundational basis for enhancing the robustness of diffusion detectors.
- We introduce a novel framework named the StealthDiffusion, which consists of Latent Adversarial Optimization strategy and Control-VAE module to refine image authenticity and reduce the spectral discrepancy.
- Extensive qualitative and quantitative experiments on large-scale diffusion datasets demonstrate the superiority of our approach in producing more indistinguishable and high-quality generated images.

## 2 RELATED WORKS

### 2.1 AI-Generated Content Stealth

The goal of *AI-Generated Content Stealth* (**AIGC-S**) task is to transform AI-generated images into forms that can evade detection algorithms without introducing visible adversarial noise. Using traditional adversarial algorithms capable of generating adversarial perturbations can misleading the target model [7, 15, 30]. However, these adversarial noises do not meet our stealth criteria.

With the advent of diffusion methods, new adversarial attack techniques have been developed that use diffusion models to create more natural-looking perturbations than traditional gradient-based methods [4, 45]. Chen *et al.* [4] manipulate the latent space of diffusion models with semantic labels to produce adversarial examples targeting the Imagenet database [8]. Similarly, Xue *et al.* [45] employ a method that iteratively adds adversarial perturbations, reconstructing them through a diffusion model at each step to create more realistic adversarial images. However, since diffusion is an AI-generated method, it might increase the chance of these images being detected by forensics detectors.

The key differences between high-quality AI-Generated Images and genuine images predominantly lie in their spectral characteristics [3, 11–13, 37]. Therefore, traditional adversarial attack methods in forgery detection have focused on reducing the spectral discrepancies between AI-generated and genuine images [10, 19, 21, 23, 26, 43]. Methods such as those proposed by [10, 19, 21] focus on the statistical differences in frequency information between AI-generated and genuine images, designing attacks based on these observations. Liu *et al.* [26] use the SRM filter [14] to extract features from AI-Generated and genuine images—features that are primarily sources of spectral differences—and train a U-Net architecture to transform the AI-Generated image features into those of genuine images. Lee *et al.* [23] employ a GAN-like architecture with a spectral discriminator to reconstruct AI-Generated images with reduced spectral discrepancies. Wu *et al.* [43] decompose AI-Generated images into high and low frequency components, adding perturbations to mislead detection methods.

To achieve more generalizable and transferable natural attacks, we explore techniques on Stable Diffusion to add adversarial perturbations while reducing spectral discrepancies with genuine images, thereby evading various image forensics detectors.

### 2.2 Image Forensics Detection

Image Forensics Detection has gained considerable attention from researchers in order to prevent the misuse of AI-generated images. Wang *et al.* [41] addressed it as a binary classification problem, training deep learning networks with fake images generated by [22] and genuine images from the LSUN dataset [46]. Recent approaches, such as [13, 38], have focused on improving detection's generalization and accuracy by extracting features from images instead of using the images themselves as the training dataset. Frank *et al.* [13] utilized the discrete Fourier transform of images for forgery detection, while Tan *et al.* [38] employed a GAN-based discriminator to convert images into gradient maps for detection. Specialized methods also exist for detecting GAN-generated fake faces [2, 29]. These studies demonstrate the effectiveness of simple supervised image forensics classifiers in detecting GAN-generated images. However, as diffusion-based generation techniques continue to advance, previous GAN-based detection methods can not adequately generalize to diffusion images. Consequently, there is growing research interest in detecting generated images produced by diffusion [33, 42], which has shown promising results in effective detection.

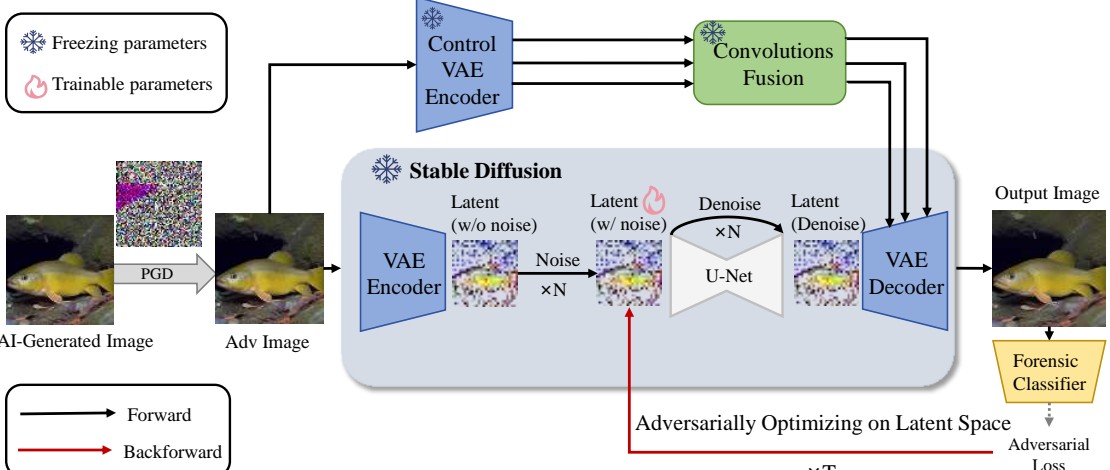

**Figure 2: Overview of our method. We introduce a small adversarial noise to the raw image using the PGD [30] method, then proceed to the Adversarially Optimizing on Latent Space step in Stable Diffusion, and the final output image is obtained by combining the outputs from the Control-VAE. This refined image will be recognized as a genuine image by the forensic detector.**

## 3 METHOD

### 3.1 Overview Framework

This section presents the overall framework of our proposed StealthDiffusion. The workflow begins by applying a Projected Gradient Descent (PGD) adversarial attack to an input image, followed by processing through a Variational Autoencoder (VAE) to extract its latent representation. Within the diffusion framework, the latent image is refined through noise addition and strategic denoising using a UNet. An adversarial loss function optimizes these features to evade detection by a surrogate classifier. To minimize spectral artifacts and reduce detectability, a Control-VAE module, trained to align the spectral frequency of the reconstructed image with genuine images, is integrated during the decoding phase via skip connections.

### 3.2 Preliminaries

Our optimization framework commences with a preprocessing phase designed to streamline the optimization challenges encountered in subsequent stages. This is particularly crucial for operations within the latent space during the stable diffusion process, which may amplify the likelihood of generated image detection. Drawing inspiration from [19], we implement the Projected Gradient Descent (PGD) technique to inject nuanced and effective adversarial perturbations into the initially generated images $x_0$. This strategic enhancement bolsters the images' ability to evade detection in the later stages of our framework. The application of these perturbations is guided by a surrogate forensic model, which is explicated in the equation that follows:

$$x_{t+1} = \text{Clip}(x_t + \eta \cdot \text{sign}(\nabla_x \mathcal{L}(S(x_t), y_{true}))), \qquad (1)$$

where $x_0$ is the initial adversarial image and $x_{t+1}$ represents its evolution after iteration $t$. The clipping function Clip ensures that

the perturbations do not exceed the imperceptibility threshold determined by $\epsilon$. $\nabla_x \mathcal{L}$ signifies the gradient of the loss function $L$, considering the true label $y_{true}$ and the surrogate forensic classifier $S$. This PGD preprocessing not only primes the images for robustness but also reduces the complexity of subsequent optimization within the diffusion process, thereby enhancing the model's ability to evade forensic detection with greater efficiency.

### 3.3 Latent Adversarial Optimization

Building on the robust foundation provided by the preprocessing stage, the adversarially perturbed image $x_{t+1}$ is transformed into a latent representation through the encoding capabilities of a Variational Autoencoder (VAE) encoder, denoted by $E$. This crucial step compresses the perturbed image into a latent format within a lower-dimensional space, optimally preparing it for the sophisticated denoising and refinement processes of the Denoising Diffusion Probabilistic Models (DDPM). The VAE encoder plays a pivotal role in this phase, distilling the essential features of the image and setting the stage for the complex operations characteristic of the subsequent DDPM-based adversarial optimization.

We then proceed to a meticulous adversarial optimization process within the latent space. This phase is vital as it exploits the inherent structural properties of the latent space to enable precise and controlled refinement of the image. Employing the strengths of Stable Diffusion Models, we conduct a series of forward and backward operations that systematically enhance the latent variables. This detailed manipulation allows us to carefully craft the adversarial features into configurations that are more resistant to forensic detection, all while maintaining the image's integrity.

the Denoising Diffusion Probabilistic Models (DDPM) employ a series of forward and reverse operations to iteratively refine the latent variables. The forward process can be mathematically represented as follows, where $z_t$ denotes the noisy latent variable at

step $t$, and $\alpha_1, ..., \alpha_N$ define a predetermined noise schedule across $N$ steps:

$$q(z_t|z_{t-1}) = \mathcal{N}(z_t; \sqrt{\frac{\alpha_t}{\alpha_{t-1}}}z_{t-1}, (1 - \frac{\alpha_t}{\alpha_{t-1}})\mathbf{I}) \quad (2)$$

This can be succinctly expressed as:

$$q(z_t|z_0) = \mathcal{N}(z_t; \sqrt{\alpha_t}z_0, (1 - \alpha_t)\mathbf{I}), \quad (3)$$

The reverse process, crucial for refining the adversarial characteristics, is defined as:

$$p_\theta(z_{t-1}|z_t) = \mathcal{N}(z_{t-1}; \mu_\theta(z_t, t), \sigma_\theta(z_t, t)) \quad (4)$$

Through $N$ iterations of these steps, the refined latent variable $z$ is then reconstructed into the final image $x'$ using the VAE Decoder $D$. To optimize the adversarial qualities of $x'$, an adversarial loss is computed using a surrogate forensic classifier $S$, with the objective of optimizing $z_N$ as shown:

$$\arg\min_{z_N} -L(S(D(z')), y_{true}), \quad (5)$$

We set the number of optimization iterations to $T$, ensuring the production of high-quality images that not only leverage the capabilities of Stable Diffusion but also remain undetectable by forensic classifiers. This strategic use of DDPM within our workflow overcomes common detection challenges, rendering the optimized images forensically robust.

### 3.4 Control-VAE Module

**Motivation Analysis.** While Latent Adversarial Optimization enhances the resistance of diffusion-generated images to detection techniques by optimizing latent variables, we identified that this optimization fails to alter the distinctive spectral signatures intrinsic to generated images. However, numerous studies [3, 12, 13, 33, 37, 41, 49] have observed significant differences between the spectral signatures of generated images and those of genuine images. These studies have identified that the spectral discrepancies primarily originate from the high-frequency components. Consequently, some research [5, 6, 35] has employed specifically designed filters to remove the content of generated images, effectively isolating most of the low-frequency components. This process yields the noise residuals of generated images, thereby providing a more intuitive demonstration of the differences in the spectral signatures between generated and genuine images. Eliminating these spectral patterns in forged images has been proven to be an important method to evade detection by recognition models [3, 10, 19, 21, 23, 26, 43]. To address and further analyze these spectral disparities, we investigated the frequency spectra produced by various diffusion methods. Specifically, for three generative methods including ADM [9] for Denoising Diffusion Probabilistic Model(DDPM), BigGAN [1] for Generative Adversarial Network(GAN), Stable_Diffusion versions 1.4 and 1.5 [36] for Latent Diffusion Model (LDM), we selected a random set of one thousand images $\{x_i\}$. we employ a commonly used denoising network [47] as our filter, which is also the filter utilized in [5, 6], to extract these noise residuals:

$$r_i = x_i - f(x_i), \quad (6)$$

We calculated the average Fourier amplitude spectra of these residuals, as visualized in Fig. 3. Our analysis indicated that images generated by the Denoising Diffusion Probabilistic Model (DDPM) tend to display spectra that closely mimic those of genuine images, with minimal detectable flaws. In contrast, the Latent Diffusion Model (LDM) spectra still exhibit a subtle grid-like pattern, characterized by high frequencies that are akin to those observed in GAN-generated images. This phenomenon could be ascribed to the decoder module's repetitive upsampling process in LDM, which inadvertently introduces high-frequency artifacts as a result of spectrum replication [3, 11, 27]. Unlike Latent Diffusion Models (LDM), traditional Denoising Diffusion Probabilistic Models (DDPM) solely utilize a Unet architecture with residual connections and do not employ a Variational Autoencoder (VAE) architecture to embed images into the latent space. Although the Unet architecture includes upsampling mechanisms, the integration of downsampling maps from the encoder with upsampling maps in the decoder through residual connections effectively mitigates artifacts introduced by upsampling. This process, as discussed in [25], robustly reduces the occurrence of such artifacts through convolutional operations that combine these features.

**Module Design.** Consequently, to mitigate the spectral discrepancies identified in the LDM-generated images, we introduce an enhanced VAE architecture embedded with residual connections and trainable convolutional layers. This innovative design not only preserves the essential characteristics of the original images but also fine-tunes the reconstruction process to produce images whose noise distributions are closely aligned with those of genuine images, as demonstrated in Fig. 4.

In pursuit of our objective, we have meticulously formulated a loss function to synchronize the noise residuals of the geneuine images with those reproduced by our model. Utilizing the DnCNN filter $f$, we calculate the noise residuals $R = \{r_i\}_{i=1,2,...,M}$ from a dataset containing $M$ genuine images $X = \{x_i\}_{i=1,2,...,M}$, as prescribed by the method delineated in Eq. 6. Applying the Discrete Two-Dimensional Fourier Transform $\mathcal{F}$ as described in Eq. 7 to these averaged residuals results in the "**Noise Prototype**", symbolized as $N_p$.

$$\hat{I}[k, l, :] = \mathcal{F}(I) = \frac{1}{HW} \sum_{x=0}^{H-1} \sum_{y=0}^{W-1} \exp^{-2\pi i \frac{x \cdot k}{H}} \exp^{-2\pi i \frac{y \cdot l}{W}} \cdot I[x, y, :], \quad (7)$$

This prototype encapsulates the aggregate noise footprint of genuine images, as corroborated by both prior research [5, 6, 35] and our spectral analysis. The calculation of $N_p$ is formalized as:

$$N_p = \mathcal{F}(\sum_{i=1}^{M} r_i), \quad (8)$$

Subsequently, our Control-VAE module processes a batch of genuine images to yield a set of reconstructed counterparts, denoted as $\{x_{b_i}^r\}$, where $b_s$ signifies the batch size. We then compute the noise residuals for this batch and apply a Fourier transform to obtain $\{N_{b_i}\}$. Our aim is to minimize the Noise Prototype Loss (**NPL**), which quantifies the discrepancy between the noise prototype and

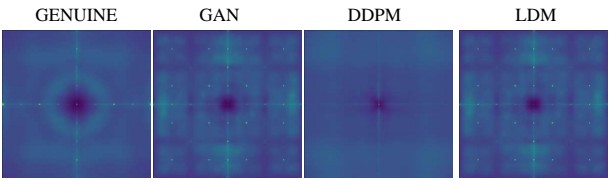

GENUINE   GAN   DDPM   LDM

**Figure 3: Fourier transform (amplitude) of the artificial fingerprint estimated from 1000 image noise residuals. First column: genuine Images from imagenet [8]. Second column: The method BigGAN [1] from Generative Adversarial Network (GAN). Third column: ADM [9] from Denoising Diffusion Probabilistic Models (DDPM). Fourth columns: Stable Diffusion (1.4 and 1.5) [36] from Latent Diffusion Model (LDM).**

the batch noise spectra, as described in Eq. 9:

$$\mathcal{L}_{NPL} = \sum_{i=1}^{b_s} \|N_p - N_{b_i}\|_2, \tag{9}$$

Inspired by [48], we configure the Convolution Fusion module with zero initialization to maintain the integrity of the original Stable Diffusion architecture. Subsequently, the meticulously trained VAE Encoder and Convolution Fusion module are integrated as independent elements within the decoder. Specifically, we denote the downscaled feature maps from the original VAE encoder at resolutions $1/2$, $1/4$, and $1/8$ as $f_1$, $f_2$, and $f_3$, respectively, and the corresponding resolution feature maps from the original VAE decoder as $g_1$, $g_2$, and $g_3$. Through Eq. 10, we fuse the feature maps from the encoder into the decoder's feature maps to obtain new feature maps $\hat{g}_1$, $\hat{g}_2$, and $\hat{g}_3$, as illustrated in Fig. 2, guiding the synthesis of the final adversarial samples. This Control-VAE process is crucial for diminishing any discernible artifacts introduced by the VAE in the Stable Diffusion process. The success and efficacy of this module are corroborated by the results of our empirical evaluations.

$$\hat{g}_i = g_i + zero\_conv(f_i) \quad i = 1, 2, 3 \tag{10}$$

To holistically optimize our module, we amalgamate the NPL with the VAE's intrinsic loss function, culminating in the composite loss equation presented in Eq. 11. Here, $\alpha$, $\beta$, and $\gamma$ represent the respective weighting coefficients for each loss component:

$$\mathcal{L} = \alpha\mathcal{L}_1 + \beta\mathcal{L}_{LPIPS} + \gamma\mathcal{L}_{NPL}, \tag{11}$$

## 4 EXPERIMENTS

### 4.1 Experimental Setups

**Dataset.** We evaluated our method on the GenImage dataset [49], which consists of 1.35 million generated images and 1.33 million genuine images from ImageNet [8]. The dataset encompasses sub-datasets generated by seven diffusion methods (ADM [9], Glide [32], Midjourney [31], Stable Diffusion 1.4 & 1.5 [36], VQDM [16], Wukong [44]), and one GAN method (BigGAN [1]). The dataset's large quantity of images and diverse generation methods allow for comprehensive analysis, making it a suitable choice for our experiments.

**Surrogate Forensic Detector.** We employed the classification evidence method proposed in [41], using EfficientNet-B0 [39], ResNet-50 [17], DeiT(Base) [40], and Swin-T(Base) [28] as backbone models.

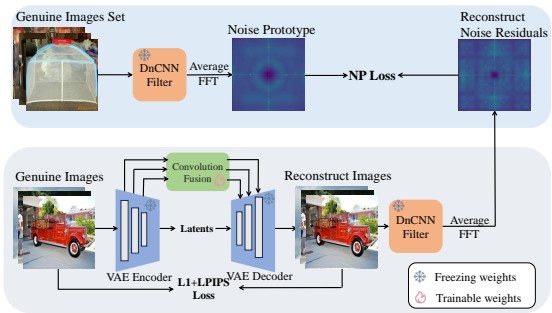

**Figure 4: The proposed Control-VAE framework extends the traditional VAE structure by incorporating a residual structure with trainable convolutions to pass the feature maps from the encoder to the decoder. While preserving the optimization loss of the traditional VAE, we additionally introduce our designed NPL loss to optimize the convolutional layers in the encoder and residual structure. This aims to reduce the distance between the noise residuals in the reconstructed images and those in the genuine images. (See Section 3.4 for more details.)**

In subsequent tables, we will abbreviate these models using their capital initials. Unlike [41], which trained only on genuine images and images generated by ProGAN [22], we trained our backbones on both generated and genuine images from GenImage, resizing all images to $224 \times 224$ and applying ImageNet normalization. We used the Adam optimizer with a learning rate of $2 \times 10^{-4}$, a batch size of 48, and trained the models for 10 epochs. The optimal weights were chosen based on the best performance on the GenImage validation set, where all backbones achieved over 98% accuracy.

**Universal Forensic Detector.** To assess the adversarial robustness of our method, we tested it against several state-of-the-art detection methods: Lgrad [38] for GAN image detection, GFF [29] and RECCE [2] for face forgery detection, and UniDetection [33] and DIRE [42] for detecting diffusion-generated images. These detection models were fine-tuned on GenImage using their original pretrained weights and achieved over 98% accuracy on its validation set.

**Baseline Attack Methods.** We compared our method with three gradient-based attack methods: FGSM [15], PGD [30], and AutoAttack (AA) [7], setting a maximum perturbation of $\epsilon = 8/255$, a pixel range of [0,1], and performing 30 iterations. We also evaluated two diffusion-based attacks, Diff-PGD [45] with 3 diffusion steps, and DiffAttack [4], which classified generated images as "AI-generated Image" and genuine ones as "Nature Image," running 10 iterations with 10 diffusion steps.

**Evaluation Metrics.** In order to comprehensively evaluate both the baseline methods and our own method, we utilized several evaluation metrics including the Attack Success Rate (ASR) as well as the image quality evaluation metrics PSNR and SSIM.

**Implementation Details.** To evaluate our attack, we resized all input images to $224 \times 224$ and randomly selected 100 images from each generation method's validation set, creating an 800-image dataset. We initiated adversarial perturbations using half the baseline value, $\epsilon = 4/255$, and set the number of iterations to 10. We

**Table 1: The performance of attack methods evaluated using the Attack Success Rate, with the first column representing the methods EfficientNet-B0(E) [39], ResNet-50(R) [17], DeiT(D) [40] used to detect adversarial samples. The second column represents different baseline attack methods FGSM [15], PGD [30], AutoAttack(AA) [7], Diff-PGD [45], DiffAttack [4], and our method. The first row represents different datasets, covering 8 sub-datasets in the GenImage dataset [49]: ADM [9], BigGAN [1], Glide [32], Midjourney [31], Stable Diffusion 1.4&1.5 [36], VQDM [16], Wukong [44]. Higher metric values indicate better performance, with the best results highlighted in bold.**

| | | ADM | BigGAN | Glide | Midjourney | SDv14 | SDv15 | VQDM | Wukong | Average |
|---|---|---|---|---|---|---|---|---|---|---|
| | FGSM | 32.00 | 38.00 | 62.00 | 89.00 | 59.00 | 64.00 | 22.00 | 46.00 | 51.50 |
| | PGD | 45.00 | 50.00 | 53.00 | 84.00 | 51.00 | 52.00 | 30.00 | 43.00 | 51.00 |
| | AA | 38.00 | 39.00 | 43.00 | 77.00 | 54.00 | 53.00 | 27.00 | 39.00 | 46.25 |
| E | DiffAttack | 7.00 | 30.00 | 17.00 | 26.00 | 14.00 | 23.00 | **53.00** | 24.00 | 24.25 |
| | DiffPGD | 34.00 | 56.00 | 12.00 | 39.00 | 44.00 | 44.00 | 29.00 | 45.00 | 37.88 |
| | Ours | **59.00** | **82.00** | **81.00** | **97.00** | **87.00** | **86.00** | 45.00 | **79.00** | **77.00** |
| | FGSM | 12.00 | 5.00 | 33.00 | 80.00 | 57.00 | 50.00 | 33.00 | 43.00 | 39.13 |
| | PGD | 14.00 | 29.00 | 52.00 | 89.00 | 77.00 | 75.00 | 38.00 | 62.00 | 54.50 |
| | AA | 10.00 | 23.00 | **55.00** | 91.00 | 81.00 | 79.00 | 39.00 | 61.00 | 54.88 |
| R | DiffAttack | 11.00 | 15.00 | 18.00 | 28.00 | 40.00 | 31.00 | 66.00 | 31.00 | 30.00 |
| | DiffPGD | **32.00** | 65.00 | 22.00 | 43.00 | 65.00 | 60.00 | 85.00 | 63.00 | 54.38 |
| | Ours | 22.00 | **41.00** | 54.00 | **95.00** | **94.00** | **97.00** | **93.00** | **90.00** | **73.25** |
| | FGSM | 18.00 | 11.00 | 35.00 | 83.00 | 32.00 | 36.00 | 12.00 | 26.00 | 31.63 |
| | PGD | 35.00 | 26.00 | 55.00 | 87.00 | 33.00 | 32.00 | 20.00 | 25.00 | 39.13 |
| | AA | 37.00 | 30.00 | 54.00 | 86.00 | 41.00 | 41.00 | 24.00 | 33.00 | 43.25 |
| D | DiffAttack | 33.00 | 25.00 | 22.00 | 30.00 | 23.00 | 22.00 | 51.00 | 21.00 | 28.38 |
| | DiffPGD | 42.00 | 42.00 | 17.00 | 29.00 | 24.00 | 24.00 | **67.00** | 20.00 | 33.13 |
| | Ours | **56.00** | **50.00** | **72.00** | **97.00** | **66.00** | **61.00** | 51.00 | **51.00** | **63.00** |

trained the Control-VAE model using genuine images from the GenImage dataset and used Stable Diffusion v2.1. The coefficients $\alpha$, $\beta$, and $\gamma$ were set at 1, 1, and 10, respectively. In the adversarial optimization phase, we applied 2 diffusion steps in the latent space with 5 iterations. To improve the diffusion algorithm's sampling speed, DDIM20 was utilized as the sampler for all diffusion methods.

## 4.2 Attack on Surrogate Forensic Detector

We conducted experiments to evaluate the effectiveness of our proposed attack method on four detectors with different backbones trained based on [41], under both white-box and black-box settings. Due to the length of the article, we only present the transfer attack success rates of Swin-T(S) [28] against other backbones: EfficientNet-B0(E) [39], ResNet-50(R) [17], and DeiT(D) [40] in Tab. 1 in the main body. Our method outperformed all other baselines in terms of average attack success rate across all datasets, demonstrating commendable transfer attack performance against various detection methods and generalization across different AI-generated methodologies. It is also noteworthy that our approach achieved an attack success rate of over 90% against the widely-used commercial AI content generation algorithm Midjourney, highlighting the advantages of our method. The complete table will be provided in the supplementary materials.

## 4.3 Attack on Universal Forensic Detector

To further demonstrate the attack capability of our method, we evaluate the transferability of attacks from the Surrogate Forensic

**Table 2: The performance of transfer attacks on Universal Forensic Detector. The first column represents the methods E [39], R [17], D [40], S [28] used to generate adversarial samples.**

| | | FGSM | PGD | AA | DiffAttack | DiffPGD | Ours |
|---|---|---|---|---|---|---|---|
| | DIRE | 74.00 | 88.50 | 86.00 | 54.00 | 72.50 | **88.50** |
| | GFF | 77.00 | 77.50 | 79.00 | 37.50 | 84.00 | **92.13** |
| E | Lgrad | 74.75 | 85.00 | 86.38 | 23.75 | 49.38 | **89.13** |
| | RECCE | 70.50 | 85.50 | 85.00 | 72.50 | 83.25 | **86.00** |
| | UniDetection | 19.25 | 16.13 | 15.38 | 13.75 | 37.00 | **46.38** |
| | DIRE | 85.00 | 92.00 | 90.00 | 58.50 | 72.00 | **96.50** |
| | GFF | 69.50 | 95.50 | 97.50 | 59.38 | 76.63 | **97.50** |
| R | Lgrad | 76.50 | 87.25 | 84.88 | 21.88 | 52.63 | **87.25** |
| | RECCE | 64.25 | 81.00 | 81.50 | 75.00 | 87.50 | **88.00** |
| | UniDetection | 16.25 | 36.63 | 39.63 | 15.50 | 23.38 | **61.13** |
| | DIRE | 77.50 | **90.00** | 89.00 | 53.00 | 61.00 | 79.00 |
| | GFF | 48.00 | 52.00 | 60.25 | 50.25 | 92.13 | **93.00** |
| D | Lgrad | 79.38 | 90.13 | 91.00 | 21.50 | 66.13 | **91.63** |
| | RECCE | 70.25 | 86.50 | 87.88 | 77.88 | 82.50 | **90.75** |
| | UniDetection | 1.75 | 4.63 | 3.13 | 11.00 | **24.63** | 18.63 |
| | DIRE | 79.50 | **98.50** | 97.50 | 58.00 | 59.50 | 95.50 |
| | GFF | 98.00 | 98.50 | 98.50 | 46.00 | 91.50 | **99.00** |
| S | Lgrad | 73.38 | 87.13 | 81.25 | 20.25 | 30.75 | **90.13** |
| | RECCE | 71.50 | 78.50 | 79.50 | 80.13 | 87.75 | **92.13** |
| | UniDetection | 6.75 | 27.13 | 24.25 | 15.13 | 25.38 | **27.75** |

Detector to the Universal Forensic Detector, specifically targeting two forensic classifiers capable of detecting images generated using

**Figure 5: Qualitative assessment of adversarial examples generated by FGSM [15], PGD [30], AutoAttack(AA) [7], DiffAttack [4], Diff-PGD [45], and our method on the GenImage dataset [49]. These samples were generated from different backbones, namely EfficientNet-B0(E) [39], ResNet-50(R) [17], DeiT(D) [40] and Swin-T(S) [28]. Although all adversarial samples successfully deceived the detectors, the adversarial samples crafted by our method exhibited a higher level of image quality.**

**Table 3: The mean of PSNR and SSIM of adversarial samples generated by different attack methods. The first column represents the methods E [39], R [17], D [40], S [28] used to generate adversarial samples.**

|   |      | FGSM  | PGD   | AA    | DiffAttack | Diff-PGD | Ours  |
|---|------|-------|-------|-------|------------|----------|-------|
| E | PSNR | 30.72 | 34.28 | **36.05** | 26.52  | 32.07    | 33.58 |
|   | SSIM | 0.73  | 0.87  | 0.87  | 0.73       | 0.88     | **0.88** |
| R | PSNR | 30.72 | 33.82 | **37.36** | 26.21  | 32.14    | 33.31 |
|   | SSIM | 0.74  | 0.86  | 0.87  | 0.74       | **0.88** | 0.87  |
| D | PSNR | 30.73 | 33.70 | **34.29** | 26.00  | 31.45    | 33.35 |
|   | SSIM | 0.76  | 0.87  | 0.87  | 0.75       | 0.87     | **0.89** |
| S | PSNR | 30.88 | 34.23 | 33.98 | 26.26      | 32.67    | **35.10** |
|   | SSIM | 0.70  | 0.86  | 0.85  | 0.75       | 0.90     | **0.91** |

**Table 4: The L2 distance of adversarial samples generated by different attack methods.**

| Method      | LDM    | FGSM   | PGD    | AA     | DiffAttack | Diff-PGD | Ours       |
|-------------|--------|--------|--------|--------|------------|----------|------------|
| L2 Distance | 0.0281 | 0.0263 | 0.0253 | 0.0257 | 0.0249     | 0.0233   | **0.0212** |

Diffusion methods [33, 42]. The results are shown in Tab. 2. Even in the black-box attack scenario, our method maintains a strong attack performance, achieving a top-two success rate compared to baseline attack methods. In particular, when utilizing ResNet50 [17], our method surpasses the second-ranked baseline attack method by 21.5% and 4.5% in terms of performance for the two detection methods.

## 4.4 Analysis

**Image Quality Analysis.** To further demonstrate the image quality of our adversarial generation method, we conducted both qualitative and quantitative analyses of the adversarial samples produced

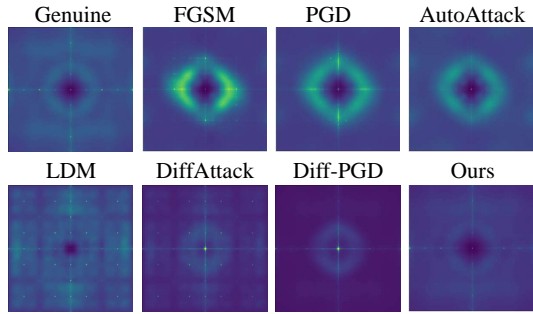

**Figure 6: Qualitative assessment of the spectral characteristics of adversarial examples generated by baseline method and our method was conducted on the GenImage dataset [49]. The term "Genuine" refers to the spectral representation of genuine images from the GenImage dataset, while "LDM" denotes the spectral representation of images generated by stable diffusion in GenImage dataset.**

by the baseline attack method and our proposed method. Fig. 5 presents the generated adversarial samples. It is evident that traditional gradient-based transfer attack methods [7, 15, 30] introduce visible noise patterns, whereas our diffusion model-based attack method produces images without noticeable noise patterns. To emphasize this observation, we extracted and enlarged a section of the image to showcase the attack results. Tab. 3 reports quantitative results for various generation methods, including the mean of PSNR and SSIM. It is evident that our method outperforms other diffusion-based attack methods in terms of image quality, achieving the highest SSIM metric score.

**Image Spectral Analysis.** To further analyze the spectral characteristics of different attack methods, we conducted both qualitative and quantitative analyses of the spectral properties of adversarial samples produced by baseline attack methods and our proposed

Genuine Image    Raw VAE(w/o PGD)    Raw VAE(w PGD)    Control VAE(w/o NPL)    Control VAE(w NPL)

**Figure 7: Fourier transform (amplitude) of the artificial fingerprint estimated from 1000 image noise residuals reconstructed using different architectures.**

method. It can be clearly observed in Fig. 6 that among the many attack strategies, our method most closely approximates the spectral signature of genuine images. In contrast, attacks based on diffusion typically carry distinctive spectral traces of the diffusion process, while gradient-based attacks introduce excessive perturbations resulting in unnatural spectral features. In Tab. 4, we quantitatively demonstrate the spectral discrepancies between adversarial samples and genuine images using the L2 distance metric. Our method outperforms the others, achieving the smallest L2 distance.

## 4.5 Ablation Study

In this section, we will conduct a series of ablation experiments on the proposed attack method.

**Core Component Analysis.** Here, we only generate attacks using Swin-T [28], while the results of the remaining backbones will be presented in the supplementary materials. Tab. 5 presents the results of different variants of our method. For the baseline method, the Control-VAE and Latent Adversarial Optimization(LAO) methods are not used. Using either Control-VAE or LAO methods yields a positive effect on the ASR metrics for both backbones, and combining the two modules can bring more performance gains. Taking the attacks generated by ResNet50 [17] as an example, the adoption of Control-VAE and LAO achieved success rate improvements of 42.5%, 11.12%, and 10.57% on Efficientnet-B0 [39], DeiT [40], and Swin-T [28] detection models, respectively. Moreover, the preprocessing stage achieves a degree of adversarial robustness by introducing minuscule adversarial noise. Our approach can further augment the success rate of transfer attacks, while the omission of this initialization phase results in a discernible performance degradation. This outcome aligns with our initial rationale for implementing such an initialization.

**Table 5: Ablation study for core components of our method. The horizontal E [39], R [17], D [40], S [28] are used to detect adversarial samples.**

| Preprocess | C-VAE | LAO | E | R | D | S |
|---|---|---|---|---|---|---|
| ✓ | ✗ | ✗ | 46.63 | 52.50 | 39.25 | 100.00 |
| ✓ | ✓ | ✗ | 67.25 | 62.75 | 51.50 | 97.13 |
| ✓ | ✗ | ✓ | 75.25 | 70.50 | 59.50 | **100.00** |
| ✗ | ✓ | ✓ | 63.25 | 68.63 | 57.13 | 100.00 |
| ✓ | ✓ | ✓ | **77.00** | **73.25** | **63.00** | 98.13 |

**Effect of Noise Prototype Loss in Control-VAE.** In Tab. 6, we compared the use of NP Loss and non-use of NP Loss in the Control-VAE module in ASR. Adding NPL to the Control-VAE achieves better performance with respect to all the metrics. Additionally, we further studied the impact of VAE on the reconstruction of genuine

**Table 6: Ablation study for NPL in Control-VAE. The first column represents the methods E [39], R [17], D [40], S [28] used to generate adversarial samples and the first row represents the methods E [39], R [17], D [40], S [28] used to detect adversarial samples.**

| | | E | R | D | S |
|---|---|---|---|---|---|
| E | w/o NPL | 100.00 | 88.13 | 63.63 | 82.25 |
| | w NPL | **100.00** | **89.50** | **66.38** | **84.13** |
| R | w/o NPL | 94.75 | 100.00 | 66.00 | 92.50 |
| | w NPL | **94.75** | **100.00** | **67.25** | **95.25** |
| D | w/o NPL | 89.13 | 88.25 | 100.00 | 96.38 |
| | w NPL | **90.75** | **89.88** | **100.00** | **97.88** |
| S | w/o NPL | 76.13 | 72.50 | 62.88 | 98.00 |
| | w NPL | **77.00** | **73.25** | **63.00** | **98.13** |

images. We extracted 1000 genuine images from the dataset and reconstructed them using Raw VAE, Control-VAE (w/o NPL), and Control-VAE (w NPL), then checked them using Efficient-B0 [39], ResNet50 [17], DeiT [40], Swin-T [28]. In Tab. 7, we demonstrate the probability of these reconstructed images being detected as genuine. The results show that using Control-VAE and NPL can minimize the probability of reconstructed genuine images being detected as generated as much as possible. We show the noise residual spectrum of these reconstructed images in Fig. 7. It also indicates that while introducing adversarial losses and raw VAE cannot reconstruct adversarial images with spectra close to genuine images, the method of using Control-VAE can effectively achieve this.

**Table 7: The probability of genuine images reconstructed using different architectures being detected as genuine by the forensic detector. The first column represents the methods E [39], R [17], D [40], S [28] used to detect reconstructed samples.**

| | E | R | D | S |
|---|---|---|---|---|
| Genuine | 100.00 | 99.40 | 100.00 | 100.00 |
| Raw VAE | 85.00 | 83.10 | 92.30 | 78.60 |
| Control-VAE(w/o NPL) | 89.80 | 87.90 | 94.40 | 81.10 |
| Control-VAE(w NPL) | **93.00** | **97.20** | **96.70** | **89.20** |

## 5 CONCLUSION

The paper proposes a framework called StealthDiffusion to enhance the robustness of diffusion model-generated images in forensic detection. StealthDiffusion adds adversarial perturbations on the latent space of stable diffusion to generate high-quality synthetic images that are resistant to detection. To further reduce the spectral differences between genuine and generated images, we introduce the Control-VAE module to improve the effectiveness of the attack. Experimental evaluations on different forensic detectors demonstrate the success and superiority of the proposed attack method compared to baseline methods.

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
