# OpenReview forum: "StealthDiffusion: Towards Evading Diffusion Forensic Detection through Diffusion Model"
_acmmm.org/ACMMM/2024/Conference — MM2024 Poster_

### Official Review · Reviewer_uRxP · 2024-05-17

**Rating:** 4
**Confidence:** 4

**Summary:**

The paper proposed a novel framework to add adversarial perturbations to the generated image to surpass the forensic detection models in identifying its non-genuine nature. Particularly, the author first optimizes the latent space of the generated image to achieve the attack goal, then adds an extra module to align the spectral space of the generated image with the genuine one to surpass the detection methods based on spectral differences. The experiment shows the effectiveness of the method. The ablation studies have well demonstrated the contribution of the work.

**Strengths:**

1.	Novelty: the proposed method introduces a control VAE module optimized by NPL, which is proved to be effective.
2.	Impressive Performances: the performance in terms of ASR is better than all the other existing works, and it retains its superiority when it comes to transferability.
3.	Good writing, easy to follow.

**Limitations:**

1. I see no defenses against the adversarial example itself to be evaluated in the paper, making the robustness of the attack still covered in the purdah of suspicion. I wonder whether some simple pre-process strategies, e.g., JPEG compress, Gaussian Noise, as well as some adaptive approaches, e.g., adversarial training, is able to mitigate the attack. Such experiments can make your work more compact.

2. Besides passive detection methods, there are also some proactive detection based on watermarking techniques. I am also interested in the related experiments and discussion.

**Suitability:**

2

---

### Official Review · Reviewer_oeTm · 2024-05-25

**Rating:** 4
**Confidence:** 4

**Summary:**

The paper aims to create AI-generated images that can evade forensic detectors and human inspection.
The paper proposes StealthDiffusion, a framework based on stable diffusion that modifies AI-generated images into high-quality, imperceptible examples of confrontations capable of evading the most advanced forensic detectors.

**Strengths:**

1. This paper is the first to focus on detectability of diffusion-generated forged images, thus laying the foundation for enhancing the robustness of diffusion detectors.
2. The new framework it proposes, called StealthDiffusion, is interesting and novel.
3. The proposed method is demonstrated by extensive qualitative and quantitative experiments on large-scale diffusion datasets.

**Limitations:**

1. In the field of deepfake detection, there are many methods to attack detectors, but the authors have ignored the experimental comparison. I think more experiments should be conducted to compare with them. These methods are:
[1] Adversarial deepfakes: Evaluating vulnerability of deepfake detectors to adversarial examples. WACV 2021
[2]  Exploring adversarial fake images on face manifold. CVPR 2021
[3] Evading deepfake-image detectors with white-and black-box attacks. CVPR 2020
[4] Adversarial threats to deepfake detection: A practical perspective. CVPR 2021
[5] Evading DeepFake Detectors via Adversarial Statistical Consistency. CVPR 2023
[6] Exploring frequency adversarial attacks for face forgery detection. CVPR 2022
2. This paper lacks in-depth discussion and comparison of similar deepfake detection fields, and ignores some important papers:
[1] UIA-ViT: Unsupervised inconsistency-aware method based on vision transformer for face forgery detection. ECCV 2022
[2] Hierarchical frequency-assisted interactive networks for face manipulation detection. TIFS 2022
[3] F2Trans: High-Frequency Fine-Grained Transformer for Face Forgery Detection. TIFS 2023

**Suitability:**

2

---

### Official Review · Reviewer_j7bp · 2024-05-26

**Rating:** 3
**Confidence:** 2

**Summary:**

This paper proposed a diffusion model-based mechanism, aiming to generate adversarial examples with high visual quality and high transferability. The key part of this paper is optimizing the latent features of the diffusion model instead of optimizing the pixel value thus reducing the visual distortion by the diffusion process. Besides, by introducing a so-called ``convolutions fusion'' mechanism, the artifacts in Fourier domain are also mitigated. Experimental results shows the effectiveness of the proposed methods.

**Strengths:**

1. The paper is easy to follow.
2. The figure in this paper well illustrates the framework.
3. Utilizing a convolutions fusion mechanism in VAE is a useful part of mitigating the artifacts.
4. The experiments show the advantages and improvements.

**Limitations:**

The idea of optimizing in the latent domain is not novel anymore as many works have already utilized such a technique as an effective way to improve the visual quality of the images (DiffAttack-2023). Based on this knowledge, the main novelty here is just the convolutions fusion mechanism, however, the training process of the convolutions fusion mechanism is not well illustrated. Therefore, I think major revisions should be made to this version. Detailed concerns need to be addressed:

1. To train the convolutions fusion network, what does the L_{LPIPS} and L_1 mean?

2. In Eq. 8, shall N_p the average value? Otherwise, N_p shall be a set.

3. In adversarial image generation, why PGD is utilized? Can the benign image be utilized to initialize the starting point of the stable diffusion latent? Since the optimization is based on the latent domain, why does PGD need to be applied first in the pixel domain?

4. What will happen if the initial benign image is a natural image instead of AI-Generated image? Will the performance still be good under the settings of \epsilon=4/255? If the input can only be the generated image, why not directly generate an adversarial image but to optimize an adversarial image?

**Suitability:**

2

---

### Official Review · Reviewer_4aKi · 2024-05-27

**Rating:** 3
**Confidence:** 4

**Summary:**

This paper proposed a framework to modify AI-generated images into adversarial examples to evade the detection of forensics detectors. The proposed framework consist of two main components, Latent Adversarial Optimization and Control-VAE, and experiments demonstrate that the effectiveness of proposed method in both white-box and black-box settings.

**Strengths:**

1. This paper specific design the LAO and Control-VAE modules, which refine image authenticity and reduce the spectral discrepancy.
2. The visual quality of the Adv image has been significantly improved.

**Limitations:**

1. It seems that the adversarial noises are directly added on the raw images, what if added on compressed ones since JPEG images are more commonly used in real world.
2. The authors demonstrate that the proposed method can reduce the spectral discrepancy, however, the artifacts in other domain should also be considered such as phase spectrum.
3. The authors used four detector(E, R, D, S) for experiments, which are all baseline network, it is better to verify on more SOTA detector.
4. The robustness of the method needs to be further discussed. The robustness discussed by the author in the article is more like a generalization than a robustness.

**Suitability:**

2

---

### Meta-Review · Area_Chair_sW2N · 2024-06-28

**Recommendation:** Accept (Poster)
**Confidence:** 5

**Metareview:**

We have received feedback from four reviewers, with three positive and one negative opinion. From the review comments, it is evident that this article has innovation and addresses the reviewers' questions well. Therefore, we agree to accept this article.